# Results of a pilot feasibility randomised controlled trial exploring the use of an electronic patient-reported outcome measure in the management of UK patients with advanced chronic kidney disease

Derek Kyte [1,2] Nicola Anderson,[2,3] Jon Bishop,[4] Andrew Bissell,[5] Elizabeth Brettell,[4] Melanie Calvert [2,6,7,8,9] Marie Chadburn [4] Paul Cockwell,[3] Mary Dutton,[3] Helen Eddington,[3] Elliot Forster,[3] Gabby Hadley,[3] Natalie J Ives,[4] Louise J Jackson [10] Sonia O'Brien,[5] Gary Price,[5] Keeley Sharpe,[5] Stephanie Stringer,[3] Rav Verdi,[5] Judi Waters,[5] Adrian Wilcockson[4]

For numbered affiliations see end of article.

**Correspondence to**
Dr Derek Kyte;
d.kyte@worc.ac.uk

## ABSTRACT

**Objectives** The use of routine remote follow-up of patients with chronic kidney disease (CKD) is increasing exponentially. It has been suggested that online electronic patient-reported outcome measures (ePROMs) could be used in parallel, to facilitate real-time symptom monitoring aimed at improving outcomes. We tested the feasibility of this approach in a pilot trial of ePROM symptom monitoring versus usual care in patients with advanced CKD not on dialysis.

**Design** A 12-month, parallel, pilot randomised controlled trial (RCT) and qualitative substudy.

**Setting and participants** Queen Elizabeth Hospital Birmingham, UK. Adult patients with advanced CKD (estimated glomerular filtration rate ≥6 and ≤15 mL/min/1.73 m$^2$, or a projected risk of progression to kidney failure within 2 years ≥20%).

**Intervention** Monthly online ePROM symptom reporting, including automated feedback of tailored self-management advice and triggered clinical notifications in the advent of severe symptoms. Real-time ePROM data were made available to the clinical team via the electronic medical record.

**Outcomes** Feasibility (recruitment and retention rates, and acceptability/adherence to the ePROM intervention). Health-related quality of life, clinical data (eg, measures of kidney function, kidney failure, hospitalisation, death) and healthcare utilisation.

**Results** 52 patients were randomised (31% of approached). Case report form returns were high (99.5%), as was retention (96%). Overall, 73% of expected ePROM questionnaires were received. Intervention adherence was high beyond 90 days (74%) and 180 days (65%); but dropped beyond 270 days (46%). Qualitative interviews supported proof of concept and intervention acceptability, but highlighted necessary changes aimed at enhancing overall functionality/scalability of the ePROM system.

**Limitations** Small sample size.

## Strengths and limitations of this study

► This is the first study to examine the feasibility of a clinical trial of electronic patient-reported outcome measures (ePROM) use in a UK chronic kidney disease (CKD) population.

► Development of the study design was overseen by a patient advisory group, which included people with lived experience of CKD.

► The ePROM intervention was configured to allow real-time integration of participant's symptom data within the electronic medical record.

► As this was a pilot study, no inferences can be made about the intervention's therapeutic efficacy.

► Our findings will help guide the design of a future randomised controlled trial aimed at exploring efficacy and cost-effectiveness.

**Conclusions** This pilot trial demonstrates that patients are willing to be randomised to a trial assessing ePROM symptom monitoring. The intervention was considered acceptable; though measures to improve longer-term engagement are needed. A full-scale RCT is considered feasible.

**Trial registration number** ISRCTN12669006 and the UK NIHR Portfolio (CPMS ID: 36497).

## BACKGROUND

Patients with advanced chronic kidney disease (CKD) commonly have a high symptom burden; increasingly so as they progress towards kidney failure.[1 2] Uncontrolled symptomology can be a particular source of

anxiety and can have a detrimental impact on patient's health-related quality of life and outcomes.[1–3]

Timely detection of symptomatic deterioration is a key component of effective disease management during this period.[3] It can be challenging, however, to identify an unexpected decline in kidney function between scheduled clinic appointments, unless a patient self-refers. Unfortunately, some patients self-refer too late because they have difficulty identifying the point at which they may require assistance. Without prompt recognition of advanced symptoms, such patients are at high risk of severe illness, emergency hospitalisation, progression to unplanned kidney replacement therapy and significantly poorer long-term outcomes, including increased mortality.[4–6]

Routine systematic capture of symptom data using electronic patient-reported outcome measure (ePROM) measures has been suggested as a low-cost method of supporting symptom monitoring and control.[7] ePROM platforms provide patients with access to short online questionnaires that allow them to share self-reported symptom data with their clinical team, often in real time, to help guide care.[8] Systems may be configured to provide patients with tailored self-management advice and to trigger clinical notifications in the advent of sudden deterioration and/or severe symptomology.[9–11]

In studies involving patients with cancer, ePROM symptom monitoring is associated with enhanced patient–clinician communication; improved patient education and self-efficacy; better symptom control; earlier detection of adverse events; improved patient quality of life; reduced use of accident and emergency services; fewer inpatient hospital episodes; and improved survival; even for 'computer-inexperienced' patients.[9–17]

The efficacy of ePROM symptom monitoring for patients with advanced CKD, has not been investigated within a randomised controlled trial (RCT); nor has the feasibility of undertaking such a trial been established. This single-centre pilot study aimed to assess the feasibility of undertaking a RCT investigating the use of monthly ePROM reporting compared with usual care in patients with advanced CKD not on dialysis.

## METHODS
### Reporting
This study is reported in accordance with the Consolidated Standards of Reporting Trials checklist for reporting a pilot/feasibility trial.[18]

### Study design
RePROM (Renal ePROM) was a single-centre, open-label, two-arm randomised controlled pilot/feasibility trial and qualitative substudy. The trial was registered with ISRCTN (ISRCTN12669006) and the UK NIHR Portfolio (CPMS ID: 36497); and the protocol has been published.[19]

### Study changes
Owing to changes in clinical practice at the host research site, made in response to the COVID-19 pandemic,

the study received approval from the Health Research Authority for early closure of follow-up (2 April 2020). This meant that follow-up was truncated for some participants and that recruitment of healthcare professionals (HCPs) to the qualitative substudy had to be suspended.

### Study setting
The trial was undertaken within the Birmingham Clinical Trials Unit (BCTU) and Centre for Patient-Reported Outcomes Research at the University of Birmingham and the Queen Elizabeth Hospital Birmingham (QEHB) within the UK National Health Service (NHS) University Hospitals Birmingham Foundation Trust.

### Patient and public involvement
Development of the study design was informed by a series of meetings held with our Patient Advisory Group (AB, SO'B, GP, KS, RV and JW), established in 2016, which included people with lived experience of CKD. Members were also involved in the ePROM intervention codesign group[20] and trial management group.

### Study oversight
An independent steering committee was convened to provide guidance to the trial management group and to review feasibility data during the trial.

### Study population
Eligible participants were adult (≥18 years old) patients under the care of the kidney services at QEHB, who met the trial definition of advanced CKD (estimated glomerular filtration rate (eGFR) ≥6 and ≤15 mL/min/1.73 m$^2$, or a projected risk of progression to kidney failure within 2 years ≥20% using the four-variable Tangri renal risk equation[21]). Participants were excluded if they met any of the following criteria: patients unwilling to use the ePROM intervention; patients who, in the opinion of the consenting professional, could not speak, read or write English sufficiently well to complete the ePROM unaided; an episode of acute kidney injury (defined in accordance with international guidelines)[22] within the last 3 months; patients meeting the trial definition of kidney failure (receiving dialysis, or scheduled to start, in the next 2 weeks, had received (or had a scheduled date to receive) a kidney transplant; or an eGFR ≤5 mL/min/1.73 m$^2$); patients with a terminal illness that, in the opinion of the clinician assessing eligibility, was likely to lead to the death of the patient within 6 months of starting participation in the study.

### Recruitment and randomisation
Members of the kidney research team at QEHB screened for potentially eligible study participants using the inclusion/exclusion criteria. Those considered eligible were provided with a patient information sheet and given the opportunity to consider participation. For patients wishing to take part in the pilot trial (and optional qualitative substudy), consent, enrolment and baseline data collection was conducted face to face in clinic. Randomisation

was provided via a web-based system developed by BCTU. Participants were randomised at the level of the individual in a 1:1 ratio to usual care (control arm) or usual care supplemented with monthly online symptom reporting using the ePROM system (experimental arm). Minimisation was used to achieve balance between: 2-year risk of progression to kidney failure (<40%, vs ≥40%, based on the four-variable Tangri renal risk equation[21]); self-reported computer experience (regular use of a computer, tablet or smartphone at least weekly, vs less than weekly); and patient-reported ethnicity ('white' vs 'non-white').

### Intervention

Participants allocated to the ePROM intervention arm were asked to complete and submit monthly symptom questionnaires using an online system and received an automated reminder to do so. In addition, patients were allowed to submit any number of additional 'ad hoc' questionnaires at any time outside of the scheduled monthly reporting dates. Development and functionality of the ePROM system has been described in detail elsewhere.[20] In summary, on questionnaire submission, automated self-management advice was provided to patients based on their responses; questionnaire data was integrated into the QEHB electronic medical record and made available to HCPs in real time; and a system algorithm triggered an automated notification which was sent to both the patient and the clinical team in the event of a severe and current symptom report. Participants allocated to the control arm received usual care. It was not possible to blind clinicians or participants due to the nature of the intervention.

### Outcomes

As this was a pilot trial there was no single primary outcome measure. The primary aims of the study were to pilot the trial protocol and assess the feasibility of undertaking a full-scale RCT exploring the use of ePROMs in the management of advanced CKD. The feasibility outcomes included the following: the proportion of eligible participants approached to take part in the trial; the proportion of eligible participants who took part in the trial; recruitment rate: the proportion of participants randomised/screened; the proportion of participants randomised/approached; the proportion of participants who completed the trial (retention); and the proportion of participants who adhered to the ePROM intervention.

This pilot trial was not powered to detect differences in outcome measures, but provided an opportunity to ensure that there were no issues with completion of the outcome data and proposed outcome measures for the main RCT. The following outcome data were collected:

► Health-related quality of life, using the paper version of the EuroQol five-dimension, five-level, questionnaire (EQ-5D-5L). The EQ-5D-5L is a reliable/validated generic measure of health status/utility commonly used internationally in cost-effectiveness and ePROM research.[10 23]

► Clinical data, including serum creatinine, calcium, phosphate, bicarbonate, albumin, eGFR, albumin-to-creatinine ratio, blood pressure and, for participants with diabetes: glucose and glycated haemoglobin.

► Event data: progression to kidney failure, contacts with HCPs in secondary care (outpatient clinics and accident and emergency), inpatient hospitalisation, death.

► Additional healthcare resource use data was also collected at each study visit.

All data were collected at baseline and 3, 6, 9 and 12 months (assessment window ±3 weeks).

### Sample size

As this was a pilot trial, no formal sample size calculation was performed. Following recommendations for pilot studies, 30 patients or more are typically required to obtain estimates of the parameters needed for sample size estimation.[24 25] To allow for a 10% drop-out and lost to follow-up, this pilot trial aimed to recruit at least 33 participants in each group, a total of 66 participants. This would allow the recruitment and retention rates to be estimated with 95% CI maximum widths of 20% and 25%, respectively.

### Statistical analysis

Analysis of feasibility and clinical outcomes was based on all participants screened and recruited. For each binary outcome, the number and percentage are reported along with an exact binomial 95% CI. Estimates of differences between groups are presented as relative risks obtained from log-binomial regression models. These estimates were unadjusted due to the low number of observed events. For continuous outcomes, the means and 95% CIs are reported. Estimates of differences between groups are presented as differences in means adjusted for minimisation variables and, for longitudinal outcomes, the corresponding baseline values. All estimates of differences are presented with 95% CIs. No p values are reported as no hypothesis testing was performed. Analysis was conducted using SPSS software, V.26 (IBM) and SAS software, V.9.4 (SAS Institute). Participants were analysed in the intervention group to which they were randomised, and all participants were included whether or not they received the allocated intervention (intention to treat). The study dataset and statistical analysis plan are available on request.

### Qualitative substudy

The qualitative substudy aimed to explore patient and HCP thoughts/experiences regarding the RePROM trial processes and intervention. Semistructured interviews were conducted by the lead author according to predefined topic guides (online supplemental appendix), but there was sufficient scope to explore novel themes where appropriate. All interviews were digitally recorded, professionally transcribed and the transcripts anonymised. Transcript data were entered into a specialist

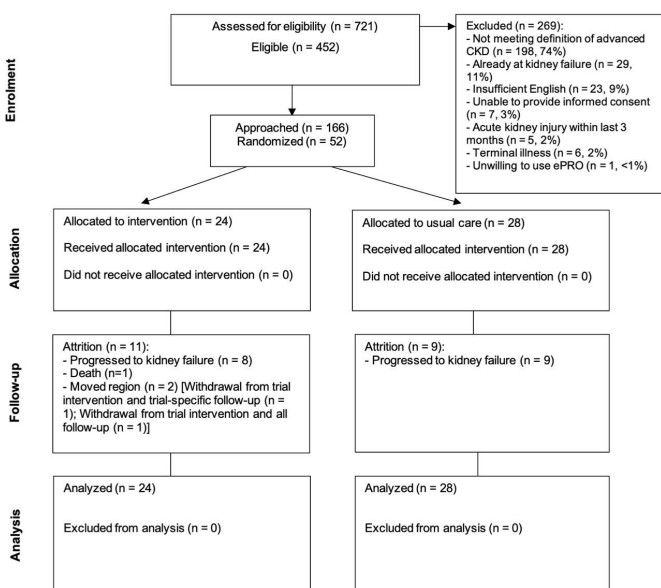

**Figure 1** Flow of participants through the trial. CKD, chronic kidney disease; ePRO, electronic patient-reported outcome.

software package (Dedoose, V.8.3.35) to aid organisation and analysis of the data. All data were analysed by the lead author using conventional content analysis.[26] Interview transcripts were examined in depth by DK, prior to first cycle coding, in which content was coded around positive and negative perceptions regarding the intervention, as well as suggested system changes.

## RESULTS
### Patients and follow-up
Recruitment was conducted at QEHB over 12 months from October 2019. The last follow-up was conducted in April 2020, which was truncated for 14 participants due to the COVID-19 pandemic. In total, 721 patients were screened, of which 452 (63%, 95% CI 59% to 66%) were eligible, and 166 were approached to take part in the trial (37% of eligible, 95% CI 32% to 41%). Fifty-two patients were randomised (figure 1) (consent rate (of approached)=31%, 95% CI 24% to 39%; consent rate (of eligible)=12%, 95% CI 9% to 15%), representing 79% of the recruitment target sample size (recruitment rate (of approached)=31%, 95% CI 24% to 39%; recruitment rate (of screened)=7%, 95% CI 5% to 9%; average monthly recruitment rate=4.3). The minimisation algorithm provided appropriate balance over 2-year risk of progression to kidney failure, however an error in the algorithm led to an imbalance in patient-reported ethnicity between groups. All participants self-reported as regular computer users.

Average follow-up was 8.0 months (SD 3.8). In total, n=2 patients withdrew from the trial during follow-up after moving geographical region (both withdrew from the intervention and one from all follow-up) (retention=96%, 95% CI 87% to 100%). During the study, n=17 patients met the trial definition of kidney failure (the

study protocol mandated exit at this point) and there was n=1 death. No patients were excluded from the analysis. Case report form return rates were excellent throughout (99.5% of all expected forms received) (online supplemental table S1).

The main reason for non-approach of screened and eligible individuals was that patients had not registered to use the existing hospital patient portal 'MyHealth' (90% of those not approached). For patients that were approached, but who were not willing to take part, reported reasons included: 'no internet access/computer inexperienced' (45%); 'not interested in research' (22%); 'too burdensome (completing ePROMs)' (11%); 'too burdensome (general)' (11%); 'issues with myHealth patient portal sign-up' (9%); 'unwell/health-related reasons' (2%); 'too burdensome (travel/trial visits)' (2%).

The average age of participants was 57 years (range 25–86), 29% were female, 37% reported 'non-white' ethnicity, 96% reported secondary level education or greater and 100% reported regular use of a computer, tablet or smartphone at least weekly. Mean baseline eGFR was 15.2, the average 2-year Tangri risk of progression to kidney failure was 43%, and the average EQ-5D index was 0.74 (table 1).

### ePROM intervention adherence and reporting patterns
Overall, 73% (95% CI 67% to 79%) of expected ePROM questionnaires were received during the trial (table 2). However, only 31% (95% CI 25% to 37%) were received within our a priori agreed compliance window (72-hours either side of the scheduled reminder date). Patients submitted 98 'ad hoc' questionnaires outside of this compliance window: an average of four per participant. Compliance over time was good, with a high proportion of participants submitting at least one scheduled questionnaire beyond 90 days postrandomisation (74%, 95% CI 52% to 90%) and after 180 days (65%, 95% CI 41% to 85%) but this proportion dropped beyond 270 days (46%, 95% CI 19% to 75%).

Patients reported 579 symptoms, the most prevalent of which included fatigue, shortness of breath, itchy/dry skin and pain (table 3, n=20 patients reported symptoms during the trial, n=4 did not report any symptoms). Most symptoms reported were mild (60%). There were 16 severe and current symptom reports (across 13 questionnaires), generated by 5 patients, representing 3% of the total number of symptoms reported across the trial (for full details around system notifications see online supplemental tables S2-S4). The symptoms driving these notifications were itchy/dry skin (37% of notifications), fatigue (25%), shortness of breath (13%), pain (13%), difficulty sleeping (6%) and ankle swelling (6%). The median time taken by staff to resolve patient notifications was 10 min (IQR 6.5–22.5) and actions included: 'telephone counselling about symptom management' (78%); and 'brought clinic appointment forwards' (22%); 'imaging/test orders' (22%); 'medication initiation/change' (11%);

| Table 1 | Baseline characteristics | | | |
|---|---|---|---|---|
| | | Monthly ePROM reports (N=24) | Usual care (N=28) | Overall (N=52) |
| **Minimisation variables** | | | | |
| Risk progression | <40% | 11 (46%) | 14 (50%) | 25 (48%) |
| | ≥40% | 13 (54%) | 14 (50%) | 27 (52%) |
| Self-reported computer experience* | 'Yes' | 24 (100%) | 28 (100%) | 52 (100%) |
| | 'No' | 0 (0%) | 0 (0%) | 0 (0%) |
| Ethnicity | 'White' | 18 (75%) | 15 (54%) | 33 (63%) |
| | 'Non-white' | 6 (25%) | 13 (46%) | 19 (37%) |
| **Demographic and other baseline variables** | | | | |
| Age, years | Mean (95% CI) | 58 (51 to 65) | 56 (50 to 61) | 57 (52 to 61) |
| Gender | Female | 7 (29%) | 8 (29%) | 15 (29%) |
| | Male | 17 (71%) | 20 (71%) | 37 (71%) |
| Highest level of education | Higher education (eg, Bachelors/Masters/Professional degree/PhD) | 9 (38%) | 9 (32%) | 18 (35%) |
| | Further education (eg, A-Levels/Vocational training) | 9 (38%) | 7 (25%) | 16 (31%) |
| | Secondary education (eg, GCSEs/O-levels) | 6 (25%) | 10 (36%) | 16 (31%) |
| | Primary education | 0 (0%) | 0 (0%) | 0 (0%) |
| | No qualifications | 0 (0%) | 2 (7%) | 2 (4%) |
| | Not known | 0 (0%) | 0 (0%) | 0 (0%) |
| Baseline medical history | Hypertension | 17 (71%) | 25 (89%) | 42 (81%) |
| | Atrial fibrillation | 1 (4%) | 1 (4%) | 2 (4%) |
| | Ischaemic heart disease | 2 (8%) | 4 (14%) | 6 (12%) |
| | Peripheral vascular disease | 0 (0%) | 3 (11%) | 3 (6%) |
| | Diabetes (type I) | 2 (8%) | 4 (14%) | 6 (12%) |
| | Diabetes (type II) | 7 (29%) | 8 (29%) | 15 (29%) |
| | Cerebrovascular disease | 0 (0%) | 0 (0%) | 0 (0%) |
| | Chronic respiratory disorder | 2 (8%) | 2 (7%) | 4 (8%) |
| | Thyroid disease | 0 (0%) | 0 (0%) | 0 (0%) |
| | Rheumatoid arthritis | 0 (0%) | 1 (4%) | 1 (2%) |
| | Anxiety/depression | 0 (0%) | 2 (7%) | 2 (4%) |
| | Cancer | 6 (25%) | 1 (4%) | 7 (13%) |
| Systolic BP (mm Hg) | Mean (95% CI) | 147.6 (139.1 to 156.0) | 146.0 (139.9 to 152.1) | 146.8 (141.7 to 151.8) |
| Diastolic BP (mm Hg) | Mean (95% CI) | 78.8 (75.2 to 82.4) | 77.4 (72.9 to 81.8) | 78.0 (75.2 to 80.9) |
| Health-related quality of life (EQ-5D-5L index) | Mean (95% CI) | 0.70 (0.60 to 0.80) | 0.78 (0.71 to 0.85) | 0.74 (0.68 to 0.80) |
| 2-year Tangri[1] risk of progression to kidney failure | Mean (95% CI) | 0.48 (0.40 to 0.57) | 0.43 (0.34 to 0.51) | 0.45 (0.39 to 0.51) |
| eGFR (mL/min/1,73 m$^2$) | Mean (95% CI) | 14.0 (12.5 to 15.6) | 15.7 (13.9 to 17.5) | 14.9 (13.7 to 16.1) |

Continued

| Table 1 | Continued | Monthly ePROM reports (N=24) | Usual care (N=28) | Overall (N=52) |
|---|---|---|---|---|
| Creatinine (µmol/L) | Mean (95% CI) | 384.0 (345.8 to 422.2) | 357.5 (316.3 to 398.8) | 369.8 (341.4 to 398.1) |
| Calcium (µmol/L) | Mean (95% CI) | 2.2 (2.2 to 2.3) | 2.3 (2.2 to 2.3) | 2.3 (2.2 to 2.3) |
| Bicarbonate (µmol/L) | Mean (95% CI) | 20.8 (19.8 to 21.9) | 21.3 (20.3 to 22.2) | 21.1 (20.4 to 21.7) |
| Phosphate (µmol/L) | Mean (95% CI) | 1.4 (1.3 to 1.5) | 1.4 (1.3 to 1.5) | 1.4 (1.3 to 1.5) |
| Albumin (g/L) | Mean (95% CI) | 40.4 (38.2 to 42.6) | 40.8 (39.0 to 42.7) | 40.6 (39.2 to 42.0) |
| ACR (mg/mmol) | Median (IQR) | 206.1 (126.9–285.2) | 178.1 (109.7–246.4) | 191.0 (139.5–242.5) |
| Blood glucose (mmol/L)† | Mean (95% CI) | 8.4 (6.8 to 9.9) | 7.0 (5.6 to 8.4) | 7.6 (6.5 to 8.6) |
| | Missing | 1 (2%) | 1 (2%) | 2 (4%) |
| HbA1c (mmol/mol)† | Mean (95% CI) | 57.2 (42.8 to 71.6) | 53.2 (44.0 to 62.5) | 54.6 (47.1 to 62.2) |
| | Missing | 4 (8%) | 3 (6%) | 7 (14%) |

*Defined as regular use of a computer, tablet or smartphone at least weekly.
†For diabetic participants.[1] Tangri N, Stevens LA, Griffith J, et al. A predictive model for progression of chronic kidney disease to kidney failure. Jama. 2011;305(15):1553–1559.[21]
ACR, albumin creatinine ratio; BP, blood pressure; eGFR, estimated glomerular filtration rate; ePROM, electronic patient-reported outcome; EQ5D-5L, EuroQol 5-Dimension, 5-Level; GCSE, General Certificate of Secondary Education; HbA1c, glycated haemoglobin.

'other' (11%), more than one type of action could be recorded for each notification (see online supplemental table S4).

### Clinical outcomes, patient-reported outcomes and healthcare utilisation

Clinical and patient-reported outcome data are available in online supplemental tables S5 and S6. As expected, there were high levels of uncertainty around all point estimates given the limited size of the sample.

Healthcare utilisation data appears in table 4. In summary, patients in the intervention arm reported 97 fewer episodes of healthcare utilisation than those in the usual care arm (mean number of episodes per patient: intervention arm=10.3, usual care arm=12.3; intervention arm 0.11 fewer mean episodes per month on trial), which included 54 fewer CKD-related specialist kidney clinic visits (mean per patient: intervention arm=5.4, usual care arm=6.5; intervention arm 0.07 fewer episodes per month on trial). Hospital inpatient stay was similar in both arms. Again, this exploratory data should be treated with caution owing to the small sample size.

### Safety, protocol deviations

There was one serious adverse event (n=1 death) reported during the trial. Two protocol deviations were recorded, 1 software error (resolved) and one informed consent form error (missing initial) (online supplemental table S7).

### Qualitative substudy

Semistructured interviews were conducted with 24 trial participants (intervention arm n=14; usual care arm n=10). Interviewee responses supported proof of concept and acceptability and indicated that the system had met our four-fold remit[20]:

1. To allow patients with advanced CKD to remotely self-report their symptoms using a simple and secure online platform.
2. To provide appropriate self-management advice to patients whose ePROM scores highlighted one or more mild/moderate/severe symptoms.
3. To allow monitoring of real-time patient ePROM symptom data and subsequent automated notification of both the patient and the clinical team in the advent of a severe symptom.
4. To incorporate longitudinal ePROM symptom data in the electronic patient record to help inform clinical consultations and support shared understanding/decision making.

A summary of qualitative findings regarding intervention positives/negatives and suggested system changes is presented in table 5. Patients highlighted benefits around login security; questionnaire structure, clarity and coverage; and felt reassurance that their questionnaire data, including their free-text comments (online supplemental table S8), were being monitored and responded to promptly and/or discussed in clinic. They also reported that the advice around symptoms and self-management was useful and helped alleviate anxiety around the symptoms they were experiencing.

The main system shortfalls, identified across the whole sample, included: failures of the reminder process meaning some patients did not receive reminder emails; a lack of clarity for some patients around which questionnaire they should complete at which time point and confusion around how to view self-management advice; difficulty navigating/scrolling through sections; occasional problems for some patients when submitting the questionnaire. Interviewees suggested a range of changes

**Table 2** ePROM compliance

| Total no of expected ePROM questionnaires* | Total received (%, 95% CI) | Total no submitted in compliance window† (%, 95% CI) | Total no of ad hoc ePROM questionnaire submissions | Mean no of ad hoc submissions per patient | No of patients on trial >90 days | Proportion of patients submitting ePROM questionnaires >90 days (95% CI) | No of patients on trial >180 days | Proportion of patients submitting ePROM questionnaires >180 days (95% CI) | No of patients on trial >270 days | Proportion of patients submitting ePROM questionnaires >270 days (95% CI) |
|---|---|---|---|---|---|---|---|---|---|---|
| 230 | 169 (73, 67 to 79) | 71 (31, 25 to 37) | 98 | 4 | 23 | 74% (52 to 90) | 20 | 65% (41 to 85) | 13 | 46% (19 to 75) |

*Accounting for questionnaire allocation date and loss to follow-up/withdrawals/death/progression to kidney failure.
†Questionnaires received within a ±72-hour time window.
ePROM, electronic patient-reported outcome measure.

aimed at addressing these shortfalls and enhancing the overall functionality of the system.

We experienced HCP recruitment challenges owing to healthcare pressures secondary to the COVID-19 pandemic. This meant that only one HCP interview was completed, precluding robust thematic analysis. We present the summary data in online supplemental table S9 for completeness.

## DISCUSSION

In this single-centre open-label randomised study, we examined the feasibility of randomising patients with advanced CKD to monthly ePROM reporting with real-time feedback of data or to usual care. We found that the majority of study indicators supported the feasibility of a full-scale RCT: patient eligibility rate (proportion of screened patients eligible) 63%; recruitment rate (of patients approached) 31%; case report form returns 99.5%; and retention 96%. In total, 52 patients were randomised (monthly recruitment rate=4.3), representing 79% of the recruitment target sample size (N=66). This level of recruitment would position the study in the top quartile of performance based on a review of recruitment and retention across 151 RCTs funded by the UK Health Technology Assessment Programme.[27] Moreover, overall adherence to the intervention was good, with patients returning 73% of expected ePROM questionnaires, although not always in the specified time windows. We have, therefore, demonstrated that it is possible to randomise and follow-up patients with high levels of data completion through to 12 months, and that an RCT is feasible.

Within our study, we found the observed pattern of ePROM reporting did not correspond with our *a priori* expectations. Relatively few patients submitted their questionnaires within our prespecified compliance window (72 hours either side of the scheduled submission date). Triangulation with qualitative data suggested that it was unlikely that this observation was related to issues around acceptability of the intervention: all participants indicated positive engagement with the system. Moreover, overall questionnaire return rates were high. A number of patients reported a failure to receive email reminders, or that emails were sent to junk folders, which may have contributed to out-of-window submissions: where patients relied on memory, rather than external prompts. Several patients suggested adding a mobile text reminder option, which they felt would be more reliable. It was our initial intention to include such an option, unfortunately, this was not possible within the existing patient portal framework. This feature will be made available as a priority within the next iteration of the system.

Our overall findings around feasibility align with similar research conducted in oncology. The feasibility of trial-based exploration of ePROM efficacy in this area has been well established and a number of trials successfully completed internationally, in the USA,[10] France[11] and in

**Table 3** ePROM intervention: reporting pattern by symptom

| | No of times reported | No of symptoms reported | | | Proportion of total symptoms reported (N=579) |
|---|---|---|---|---|---|
| | | Mild (%) | Moderate (%) | Severe (%) | |
| Fatigue | 135 | 69 (51) | 60 (44) | 6 (4) | 23% |
| Shortness of breath | 109 | 88 (81) | 17 (16) | 4 (4) | 19% |
| Itchy/dry skin | 102 | 53 (52) | 42 (41) | 7 (7) | 18% |
| Pain | 87 | 54 (62) | 29 (33) | 4 (5) | 15% |
| Lack of appetite | 57 | 35 (61) | 22 (39) | 0 (0) | 10% |
| Ankle swelling | 21 | 11 (52) | 9 (43) | 1 (5) | 4% |
| Nausea | 20 | 13 (65) | 7 (35) | 0 (0) | 3% |
| Difficulty sleeping | 17 | 7 (41) | 9 (53) | 1 (6) | 3% |
| Faintness/dizziness | 11 | 6 (55) | 5 (45) | 0 (0) | 2% |
| Restless legs or difficulty keeping legs still | 10 | 7 (70) | 3 (30) | 0 (0) | 2% |
| Diarrhoea | 10 | 5 (50) | 5 (50) | 0 (0) | 2% |
| Problems with fistula | 0 | 0 (0) | 0 (0) | 0 (0) | 0% |
| Total | 579 | 348 (60) | 208 (36) | 23 (4) | |

ePROM, electronic patient-reported outcome.

the UK.[28] Within kidney research, while the feasibility of routine collection of ePROMs in clinical practice has been supported,[29 30] there has been relatively little research around trial feasibility until recently. The 'symptom monitoring with feedback trial', is a registry-based pilot cluster RCT among Australian and New Zealand adults with end-stage kidney disease managed on haemodialysis; due for completion in 2020/2021.[31] Early findings from the pilot study suggest feasibility and acceptability when implementing ePROMs with feedback to clinicians in Australian haemodialysis centres, supporting progress to a follow-on multicentre RCT.[32]

Previous ePROM trials have commonly included a primary outcome based around health-related quality of life, for example, measured using the EQ-5D.[10] Based on our study population data, it would require a total of 348 participants to detect a clinically meaningful 0.07 reduction in EQ-5D-5L index[33] (SD=0.18, p=0.05, 90% power, adjusting for 20% attrition). This sample size appears achievable based on the successful implementation of previous UK-led kidney trials with similar (or greater) sample size requirements.[34 35]

While the study intervention was well received by patients and demonstrated proof of concept, there were a number of suggested improvements that may enhance longer-term engagement with the system, for example: simplification the interface and, in particular, improvements to the reminder functionality; incorporation of automated dietary advice; and the inclusion of additional questionnaire items around the psychological impacts associated with CKD. In addition, it was suggested that use of the intervention within a multicentre trial may necessitate system-level modifications to ensure compatibility with different IT infrastructures at other hospitals.

Work conducted within a UK oncology setting has shown that it is possible to integrate a single ePROM system across multiple NHS trusts, each with unique IT platforms, but that repeated integration at each separate site often takes considerable time and resources.[9] Our own experience of linking an ePROM to an existing hospital-based patient portal was mixed. Positives included the in-built security aspects, which some patients particularly valued, and also the ability to share data within the electronic medical record relatively easily. Negatives included functionality issues around the interface and the lack of some important features, for example, text reminders and smartphone compatibility. In addition, issues with sign-up to the patient portal for some patients meant that study staff could not approach them to take part in the trial without first arranging access to the patient portal, which created a substantial barrier to recruitment.

Looking ahead to the roll-out of an ePROM system within a multicentre trial, and also considering future potential implementation in clinical practice, the use of a single hospital patient portal as the foundation platform may hinder effective scale-up. Any ePROM system would ideally require full integration with the electronic healthcare record at each site, and also a unified interface, to maximise the likelihood of success and utility. In a recent renal stakeholder summit aimed at developing a UK ePROM roadmap—involving patients, HCPs, academics and funders/renal organisations (including the Renal Association, British Renal Society, Kidney Care UK, National Kidney Federation, Kidney Research UK)— the development of a single online ePROM gateway/dashboard was identified as a key priority.[36] Such a dashboard would provide patients with a simple and consistent point of entry and allow them the flexibility to configure

**Table 4** Summary of healthcare utilisation

| NHS service category | CKD-related | | | | Not CKD-related | | | | CKD relationship unknown | | | |
|---|---|---|---|---|---|---|---|---|---|---|---|---|
| | Intervention (N=24) | | Usual care (N=28) | | Intervention (N=24) | | Usual care (N=28) | | Intervention (N=24) | | Usual care (N=28) | |
| | Episodes | NHS hospital inpatient stay (days) | Episodes | NHS hospital inpatient stay (days) | Episodes | NHS hospital inpatient stay (days) | Episodes | NHS hospital inpatient stay (days) | Episodes | NHS hospital inpatient stay (days) | Episodes | NHS hospital inpatient stay (days) |
| GP appointment | 1 | | 4 (n=2) | | 14 (n=9) | | 23 (n=15) | | 0 | | 4 (n=2) | |
| GP out of hours service | 0 | | 0 | | 0 | | 1 | | 0 | | 0 | |
| Specialist kidney clinic | 129 (n=22) | | 183 (n=26) | | 1 | | 0 | | 0 | | 0 | |
| NHS outpatient clinic (other than specialist kidney clinic) | 10 (n=6) | | 15 (n=12) | | 41 (n=13) | | 74 (n=17) | | 1 | | 1 | |
| NHS walk-in centre | 0 | | 0 | | 1 | | 0 | | 0 | | 0 | |
| NHS 111/NHS direct telephone call | 0 | | 0 | | 1 | | 1 | | 0 | | 0 | |
| A&E | 1 | | 0 | | 2 (n=2) | | 5 (n=3) | | 1 | | 1 | |
| NHS hospital inpatient stay | 4 (n=3) | 7 | 2 (n=2) | 2 | 2 (n=2) | 7 | 2 (n=2) | 8 | 0 | | 2 (n=2) | 2 |
| Other: | 9 (n=5) | | 19 (n=13) | | 27 (n=4) | | 8 (n=3) | | 2 (n=2) | | 0 | |
| Imaging | 3 | | 6 | | 2 | | 1 | | 1 | | 0 | |
| Home visit | 2 | | 5 | | 0 | | 0 | | 0 | | 0 | |
| Phlebotomy | 1 | | 1 | | 0 | | 0 | | 0 | | 0 | |
| Health education/ roadshow/open day | 1 | | 1 | | 0 | | 0 | | 0 | | 0 | |
| Chemotherapy | 0 | | 0 | | 8 | | 0 | | 0 | | 0 | |
| Ophthalmology procedure | 0 | | 0 | | 1 | | 0 | | 0 | | 0 | |
| Other (NHS) | 2 | | 5 | | 1 | | 7 | | 1 | | 0 | |
| Other (private) | 0 | | 0 | | 15 | | 0 | | 0 | | 0 | |
| Total | 154 (n=22) | 7 | 222(n=26) | | 89 (n=14) | | 114 (n=23) | | 4 (n=2) | | 8 (n=4) | |

A&E, Accident and Emergency; GP, general practice; NHS, National Health Service.

**Table 5** Summary of qualitative findings regarding intervention positives/negatives and suggested system changes

| Theme **subtheme** | Illustrative quote(s) |
|---|---|
| **Intervention positives** | |
| Questionnaire data acted on | "On a few occasions I was very impressed that what I had put on the form, obviously had been noticed and had been picked up. And was discussed with me at clinic and I thought that was one of the big positives of the form itself."(Patient 01) |
| Provided reassurance | "…it does give you some reassurance if you can be told, well that's normal for the problems you've got."(Patient 02) |
| Quick to complete | "The first one probably took me quarter of an hour because I read through it very carefully and double checked what I was saying as I went along. But once I'd done a couple then it was sort of less than ten minutes… I sort of answered the questions as I felt at the time… But it was a breeze once I got used to it that was fine it was easy to fill in."(Patient 03) |
| Alleviated anxiety | "I found it positive. I think it takes worries away to be honest with you… You have the advice that was given, so you didn't feel as if you're the only person that ever-had itchiness before. It was obviously something that was very common. So, I would have said it alleviated any anxiety, for me."(Patient 01) |
| Questionnaire structure/ content | "I think the questions, they're quite clear and quite precise."(Patient 04); "…my symptoms… headaches, itchy skin, swelling which it covered, tiredness which it covered… I think it covered everything from my point of view."(Patient 05) |
| Provision of guidance | "…it prompted you to give the QE a ring and discuss it, you know what I mean… you know like feeling worse and feeling tired or whatever, just to ring up and speak to somebody cause sometimes you don't… you just don't do that… you just carry on, you just carry on till your next appointment. So, it made you think about it."(Patient 06) |
| Immediate clinical assistance | "…it's nice to know that, you know… if anything is going wrong then I can get help more or less straightaway."(Patient 07) |
| Free-text comments | "Initially I was filling the form in and putting very little additional information on. Latterly I was putting a lot more information on and I was very pleased on two occasions that when I went for my renal check-up, the points that I'd made had been noticed and were brought up… it was an additional form of communication in that if I'd got a concern or something was happening, I could put it on the form… and you could use it to answer questions then as to how you were coping, what you were doing and how you were feeling."(Patient 01) |
| Self-management advice | "…very useful because as a lay person not understanding the functions of the body, not that well if you see what I mean, it's useful sometimes to get a bit of guidance as to where you need to go."(Patient 03) |
| Login security | "…I think the security of, if you like, the double tier I think is very, very good indeed."(Patient 08) |
| User-friendly | "I think it's quite simple and user friendly."(Patient 04) |
| **Intervention negatives** | |
| Reminder failures | "…some of the time it didn't come through on my daughter's iPhone and then it would come through the next month but miss a month… Seemed to be hit and miss sometimes."(Patient 07) |
| Questionnaire completion | "The complicated bit, which I did struggle with, was trying to get up the latest questionnaire, which needed to be completed…"(Patient 08); "I would actually number the questionnaires so you can tell which ones you've done and completed… sometimes I didn't know which ones I'd done and which ones I hadn't done…"(Patient 05) |
| Prominence of next steps and self-management advice | "Yeah, I don't remember seeing too much of that [information] at the end of it to be honest."(Patient 15) |
| Difficulty navigating through multiple sections within the system | "…for some reason one of the sections within a section… I could scroll down but the inner bar I couldn't scroll down completely… there were like 10 questions, maybe 12 questions, and you could get down to question eight, but I couldn't get down to the last two…"(Patient 09) |
| Difficulty submitting the questionnaire | "…on two separate occasions we did try and fill it out but then the problem is there was never a finish or a continuation of the questionnaire, so we couldn't exactly finish it…"(Patient 10) |
| **Suggested system changes** | |
| Improve reminders | "…perhaps like my daughter found that, you know, it was hit and miss when the questionnaire [reminders] came through. That could be improved on…"(Patient 07) |
| Enhance/simplify interface | "…navigating your way through the electronic system… could be made a bit easier."(Patient 08) |

**Table 5** Continued

| Theme subtheme | Illustrative quote(s) |
|---|---|
| Incorporate dietary advice | "…my major one really, which I've been surprised at, was the lack of information regarding, you know, diet…"(Patient 11) |
| Incorporate questions around psychological well-being/mood | "I think just having that questionnaire to see how your mood is and how you can look back on it and see where, like, how you can improve and how you can change it slightly and try and move on from there…"(Patient 10) |
| Timing of questionnaire completion related to clinical encounter/receiving results | "I'm getting the [clinic] results sometimes before I answer the questionnaire, and I think that possibly can end in user bias 'cause if my results are not very good then sometimes that can translate into feeling bad, you know, rather than the other way round, if you know what I mean?"(Patient 12) |
| Incorporate other symptom questions | "I think it's worthwhile [adding]… leg cramps… it's just when you're in bed at night and lying down. It'll be like absolutely agonising, just like really painful… it is one of the key symptoms, yeah."(Patient 04) |
| Tick-box option to prompt contact with the clinical team | "I'd perhaps have the tick box at the end of the questions… to say 'could somebody ring you' would be a good idea… for someone to give you that reassurance with a phone call… of how to ease the symptoms."(Patient 05) |
| Simplify the questionnaire submission process | "I found a little bit of confusion on the last page where you, they showed you your answers, what you'd put, there's submit button on that page. I had to come back a page to submit it, that caused confusion a couple of times."(Patient 01) |
| Make data available to GPs | "…the GP side of things in the UK isn't necessarily that well linked into the hospital system… with the technology that we have these days you'd think that it would be sensible to have the GP on if you like a version of 'MyHealth' so they can see exactly what the hospital are seeing, obviously within the rules of confidentiality… I think the more integrated it is the better it will work"(Patient 03) |
| Combine questionnaire data with other clinical/lifestyle information collected at home | "…it was just my wondering whether there was another level perhaps… whether blood pressure something like that…things like the blood pressure and weight I have to record every day anyway…"(Patient 13) |
| Consider flexibility in setting notification thresholds for different symptoms | "Have the same system as the failsafe system but don't have it as severe. Maybe say level three, make it to level two or level one."(Patient 14) |

GP, general practitioner.

the platform to their liking, for example, around how reminders were configured/delivered, how their data and clinical advice were presented, or which primary/secondary care providers would have permissions to access their symptom information. Back-end development of application programming interfaces would then allow permitted healthcare providers to securely 'pull' appropriate data into their electronic medical record, regardless of their underlying system architecture.

### Strengths and limitations

This is the first UK study conducted in a CKD population that has explored the feasibility of ePROM capture/feedback with real-time integration within the electronic medical record. Our findings will help guide the design of a future RCT aimed at exploring efficacy and cost effectiveness. As this was a pilot study, no inferences can be made about the intervention's therapeutic efficacy. Nevertheless, clinical data around eGFR, risk of progression to kidney failure and healthcare utilisation show trends towards improvement in the intervention arm, suggesting further research is warranted.

The attrition rate for this study was larger than expected, owing to a higher proportion of patients progressing to kidney failure than anticipated (38% of patients randomised, vs 20% predicted). While this demonstrated the effectiveness of our recruitment strategy, which targeted patients with advanced CKD at risk of progression, the sample size for a future trial may need to be adjusted accordingly to account for this observation depending on the exact nature of the primary outcome.

During the qualitative process analysis, it was not possible to conduct dual-coding or triangulation, this should be taken into account when interpreting the findings.

The prespecified data analysis plan for this pilot study did not stipulate capture of the reason for starting dialysis, only the start date and type of dialysis was recorded.

Finally, a sizeable proportion of patients who were approached during study recruitment declined participation owing to concerns around internet access/computer inexperience. While, anecdotally, reports suggest that patients have become much more comfortable with

the use of digital healthcare necessitated during the COVID-19 pandemic, any future RCT should focus on broadening study accessibility and reducing the possibility of digital exclusion by: (1) ensuring the use of a simple user-friendly platform, with adequate training/support in place at the outset and (2) potentially providing an offline, for example, paper-based, PRO option.

## CONCLUSIONS

This UK single-centre, open-label, randomised controlled pilot study has demonstrated that it is feasible to conduct a trial incorporating online ePROM symptom reporting, with high rates of data completion. Based on patient feedback and system data, improvements to our ePROM intervention should be implemented to enhance functionality, long-term engagement and scalability prior to a multicentre RCT.

**Author affiliations**
[1]School of Applied Health & Community, University of Worcester, Worcester, UK
[2]Centre for Patient Reported Outcomes Research, Institute of Applied Health Research, College of Medical and Dental Sciences, University of Birmingham, Birmingham, UK
[3]University Hospitals Birmingham NHS Foundation Trust, Birmingham, UK
[4]Birmingham Clinical Trials Unit (BCTU), Institute of Applied Health Research, University of Birmingham, Birmingham, UK
[5]Patient Advisory Group, Centre for Patient-Reported Outcomes Research, Institute of Applied Health Research, University of Birmingham, Birmingham, UK
[6]Birmingham Health Partners Centre for Regulatory Science and Innovation, University of Birmingham, Birmingham, UK
[7]National Institute for Health Research (NIHR) Birmingham Biomedical Research Centre, University of Birmingham, Birmingham, UK
[8]National Institute for Health Research (NIHR) Applied Research Collaboration (ARC) West Midlands, University of Birmingham, Birmingham, UK
[9]National Institute for Health Research (NIHR) Surgical Reconstruction and Microbiology Research Centre, University Hospitals Birmingham NHS Foundation Trust and University of Birmingham, Birmingham, UK
[10]Health Economics Unit, Institute of Applied Health Research, University of Birmingham, Birmingham, UK

**Acknowledgements** The authors would like to thank all participants in the study. We would also like to thank the kidney research team and kidney care team at Queen Elizabeth Hospital Birmingham and the Birmingham Clinical Trials Unit for helping to run and deliver the trial. We would like to acknowledge Anita Walker for her administrative support and the RePROM Patient Advisory Group for their input into the design of the study. We thank all members of the Trial Steering Committee (Dr Andrew Mooney, Adult Renal Services, Lincoln Wing, St James University Hospital, Leeds, UK; Dr Kirstie Haywood, Warwick Research in Nursing, Warwick Medical School, University of Warwick, UK; Dr Mark Jesky, Department of Nephrology, Nottingham University Hospitals NHS Trust, Nottingham, UK) for their advice and support. We would also like to thank Profs Ethan Basch (University of North Carolina, United States), Niels Hjøllund (Arhuus University, Denmark) and Galina Velikova (Patient Outcomes Group, University of Leeds, United Kingdom) for their support and design input.

**Contributors** DK is the chief investigator and guarantor and takes final responsibility for study design, conduct and decision to submit for publication. DK led the study design process with input from NA, JB, AB, EB, MCa, MCh, PC, MD, HE, GH, NJI, LJJ, SO'B, GP, KS, SS, RV and JW. DK, NA, EB, MCh, PC, MD, HE, EF, GH, SS and AW were involved in the acquisition of data. DK and JB conducted the analysis with support from NJI. DK prepared the first draft of the manuscript with approval from all authors. All investigators (DK, NA, JB, AB, EB, MCa, MCh, PC, MD, HE, GH, NJI, LJJ, SO'B, GP, KS, SS, RV, JW, EF, AW) provided critical input regarding the interpretation of findings, were involved in revising the manuscript for its important intellectual content and read and approved the final manuscript.

**Funding** This paper presents independent research funded by the National Institute for Health Research (NIHR) Post-Doctoral Fellowship Scheme, grant number PDF-2016-09-009.

**Competing interests** EB, MCh, NJI and JB report grants from NIHR. MCa is an NIHR Senior Investigator and receives funding from the NIHR Birmingham Biomedical Research Centre, the NIHR Surgical Reconstruction and Microbiology Research Centre and NIHR Applied Research Collaboration West Midlands at the University of Birmingham and University Hospitals Birmingham NHS Foundation Trust, Health Data Research UK, Innovate UK (part of UK Research and Innovation), Macmillan Cancer Support, UCB Pharma and GSK. MC has received personal fees from Astellas, Takeda, Merck, Daiichi Sankyo, Glaukos, GSK and the Patient-Centered Outcomes Research Institute (PCORI) outside the submitted work. DK reports grants from Macmillan Cancer Support, Innovate UK, the NIHR, NIHR Birmingham Biomedical Research Centre, and NIHR SRMRC at the University of Birmingham and University Hospitals Birmingham NHS Foundation Trust, and personal fees from Merck outside the submitted work.

**Patient and public involvement** Patients and/or the public were involved in the design, or conduct, or reporting, or dissemination plans of this research. Refer to the Methods section for further details.

**Patient consent for publication** Not applicable.

**Ethics approval** This study was approved by the West Midlands Edgbaston Research Ethics Committee (Ref: 18/WM/0013) on 23 February 2018 (ePROM finalisation and pilot trial). Participants gave informed consent to participate in the study before taking part.

**Provenance and peer review** Not commissioned; externally peer reviewed.

**Data availability statement** Data are available on reasonable request. The datasets used and/or analysed during the current study are available from the Birmingham Clinical Trials Unit on reasonable request via the corresponding author.

**ORCID iDs**
Derek Kyte http://orcid.org/0000-0002-7679-6741
Melanie Calvert http://orcid.org/0000-0002-1856-837X
Marie Chadburn http://orcid.org/0000-0003-0635-6852
Louise J Jackson http://orcid.org/0000-0001-8492-0020

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
