## [Reviewer comments · BMJ Open]

ARTICLE DETAILS

TITLE (PROVISIONAL)	Results of a pilot feasibility randomised controlled trial exploring the use of an electronic patient-reported outcome measure in the management of UK patients with advanced chronic kidney disease.
AUTHORS	Kyte, Derek; Anderson, Nicola; Bishop, Jon; Bissell, Andrew; Brettell, Elizabeth; Calvert, Melanie; Chadburn, Marie; Cockwell, Paul; Dutton, Mary; Eddington, Helen; Forster, Elliot; Hadley, Gabby; Ives, Natalie; Jackson, Louise; O'Brien, Sonia; Price, Gary; Sharpe, Keeley; Stringer, Stephanie; Verdi, Rav; Waters, Judi; Wilcockson, Adrian

VERSION 1 – REVIEW

REVIEWER	Rosansky , Steven William Jennings Bryan Dorn Veterans Affairs Medical Center,, Research
REVIEW RETURNED	09-Jun-2021

GENERAL COMMENTS	very important pilot and area of research please include in the revised protocol the symptoms that justified the start of dialysis. this is an area of critical import since many patients start dialysis early art e GFR above 10 and the reasons why they started early are not defined.
---

REVIEWER	Flythe, Jenny University of North Carolina at Chapel Hill
REVIEW RETURNED	10-Aug-2021

GENERAL COMMENTS	Thank you for inviting me to review this manuscript reporting a feasibility study of use of an electronic patient-reported outcome measure (ePROM) in the management of patients with advanced CKD. The study is timely as there is great interest in incorporating routine PROM use in kidney care, but little setting-specific data regarding effectiveness and implementation of PROMs. This pilot study provides feasibility data for a larger randomized trial. Overall, the findings are well-reported and clear. I have two major comments and 1 minor comment for the authors.  1. The authors are appropriately cautious in their interpretation of the clinical results given the small sample size and the a priori exploratory nature of these outcomes. As the adjusted mean differences (95% CIs) cannot be interpreted, I would suggest removing them all from the manuscript, and instead, present only the raw differences without statistical testing. 2. The qualitative analysis of the interviews appears to be under-developed. The authors note that conventional content analysis was used but no detail of the coding approach is provided.
---

	Second, it appears that only 1 person coded the data. There is no mention of a second coder or triangulation activities among researchers. Furthermore, the reported themes are not traditional themes but instead categorical observations. Related, there is only 1 provider (acknowledged by the authors), Data from more than 1 provider would be needed to perform a rigorous qualitative analysis. I would suggest dropping the provider data or presenting it separately as anecdotal since thematic analysis of a single interview is not possible. 3. Finally, consider decreasing the size of the tables. Removing the columns noted in comment #1 will help with this, but they could likely be further pared down to make them more digestible for the reader. For example, consider moving the laboratory-related findings to the supplement.
--	--

VERSION 1 – AUTHOR RESPONSE

Reviewer 1: please include in the revised protocol the symptoms that justified the start of dialysis. this is an area of critical import since many patients start dialysis early at e GFR above 10 and the reasons why they started early are not defined.

Author response: Thank you for this comment. The pre-specified data analysis plan for this pilot study did not stipulate capture of the reason for starting dialysis, only the start date and type of dialysis was recorded (this has been added as a limitation on page 25). Unfortunately, we are therefore unable to add this information. However, we agree that this would indeed be useful data to collect in the main RCT and we therefore intend to capture the appropriate information in the relevant case report form.

Reviewer 2: The authors are appropriately cautious in their interpretation of the clinical results given the small sample size and the a priori exploratory nature of these outcomes. As the adjusted mean differences (95% CIs) cannot be interpreted, I would suggest removing them all from the manuscript, and instead, present only the raw differences without statistical testing.

Reviewer 2: ...consider decreasing the size of the tables. Removing the columns noted in comment #1 will help with this, but they could likely be further pared down to make them more digestible for the reader. For example, consider moving the laboratory-related findings to the supplement.

Author response: Thank you for these comments. We have moved 2 of the larger tables ('Numeric outcome measures by trial arm and data collection point' And 'Binary outcome measures by trial arm and data collection point') to the supplementary appendix. The former of these does still include the adjusted mean differences, owing to the fact that this was pre-specified in our statistical analysis plan. But we agree, it is more appropriate to include this information in the appendix given the level of imprecision around the estimates. In addition, the size of Table 7 ('Summary of qualitative findings regarding intervention positives/negatives and suggested system changes') has been significantly reduced in size – see below.

Reviewer 2: The qualitative analysis of the interviews appears to be under-developed. The authors note that conventional content analysis was used but no detail of the coding approach is provided. Second, it appears that only 1 person coded the data. There is no mention of a second coder or triangulation activities among researchers. Furthermore, the reported themes are not traditional themes but instead categorical observations. Related, there is only 1 provider (acknowledged by the

authors), Data from more than 1 provider would be needed to perform a rigorous qualitative analysis. I would suggest dropping the provider data or presenting it separately as anecdotal since thematic analysis of a single interview is not possible.

Author response: Thank you for this comment. We have added more detail around the qualitative analysis methods in the 'Qualitative sub-study' section. Unfortunately, owing to resource constraints, it was not possible to conduct dual-coding or triangulation of the qualitative data. This has now been added as a limitation on page 25. We have outlined the issues regarding recruitment of HCPs and the resulting inability to conduct rigorous qualitative analysis. Instead, we now present this particular summary data in the supplementary appendix (Table S9) for completeness.